# Hierarchical Harmonization of Atom-Resolved Metabolic Reactions across Metabolic Databases

**DOI:** 10.3390/metabo11070431

**Published:** 2021-06-30

**Authors:** Huan Jin, Hunter N. B. Moseley

**Affiliations:** 1Department of Toxicology and Cancer Biology, University of Kentucky, Lexington, KY 40536, USA; huan.jin@uky.edu; 2Department of Molecular & Cellular Biochemistry, University of Kentucky, Lexington, KY 40536, USA; 3Markey Cancer Center, University of Kentucky, Lexington, KY 40536, USA; 4Superfund Research Center, University of Kentucky, Lexington, KY 40506, USA; 5Institute for Biomedical Informatics, University of Kentucky, Lexington, KY 40536, USA

**Keywords:** metabolite, compound harmonization, reaction harmonization, metabolic network, metabolic model, subgraph isomorphism

## Abstract

Metabolic models have been proven to be useful tools in system biology and have been successfully applied to various research fields in a wide range of organisms. A relatively complete metabolic network is a prerequisite for deriving reliable metabolic models. The first step in constructing metabolic network is to harmonize compounds and reactions across different metabolic databases. However, effectively integrating data from various sources still remains a big challenge. Incomplete and inconsistent atomistic details in compound representations across databases is a very important limiting factor. Here, we optimized a subgraph isomorphism detection algorithm to validate generic compound pairs. Moreover, we defined a set of harmonization relationship types between compounds to deal with inconsistent chemical details while successfully capturing atom-level characteristics, enabling a more complete enabling compound harmonization across metabolic databases. In total, 15,704 compound pairs across KEGG (Kyoto Encyclopedia of Genes and Genomes) and MetaCyc databases were detected. Furthermore, utilizing the classification of compound pairs and EC (Enzyme Commission) numbers of reactions, we established hierarchical relationships between metabolic reactions, enabling the harmonization of 3856 reaction pairs. In addition, we created and used atom-specific identifiers to evaluate the consistency of atom mappings within and between harmonized reactions, detecting some consistency issues between the reaction and compound descriptions in these metabolic databases.

## 1. Introduction

Metabolic models describe the inter-conversion of metabolites via biochemical reactions catalyzed by enzymes, providing snapshots of the metabolism under a given genetic or environmental condition [1,2]. Metabolic models of metabolism have proven to be an important tool in studying systems biology and have been successfully applied to various research fields, ranging from metabolic engineering to system medicine [3,4,5,6,7]. Advances in analytical methodologies like mass spectroscopy and nuclear magnetic resonance greatly improve the high-throughput detection of thousands of metabolites, enabling the generation of large volumes of high-quality metabolomics datasets [8,9] that greatly facilitate metabolic research. As a next major step, incorporating reaction atom-mappings into metabolic models enables metabolic flux analysis of isotope-labeled metabolomics datasets [10,11,12,13], which will contribute to the large-scale characterization of metabolic flux molecular phenotypes and prediction of potential targets for gene manipulation [4]. Building reliable metabolic models heavily depend on the completeness of metabolic network databases. However, a relatively complete metabolic network, especially at an atom-resolved level, is practically not available [14].

Therefore, to construct an atom-resolved metabolic network, the very first major step is to integrate metabolic data from various metabolic databases without redundancy [15], which remains extreme labor-intensive. This is partially due to problems in the individual databases [16]. Common issues include non-unique compound identifiers, reactions with unbalanced atomic species, and enzyme catalyzing more than one reaction [17]. Moreover, incompatibilities of data representations (like compound identifiers) and incomplete atomistic details (like the presence of R groups and lack of atom and bond stereochemistry) across databases are key bottlenecks for the rapid construction of high-quality metabolic networks [18]. Great efforts have been made to map different compound identifiers across metabolic databases [19,20]. Some algorithms use logistic regression to compute the similarity between strings generated by concatenating a variety of compound features, which requires careful selection of compound features that can well characterize a string pair by capturing the similarity between different variations as well as underlining the difference between descriptions which are not synonymous [6]. Alternatively, utilization of unique chemical identifier independent from a particular database, like InChI [21,22] or SMILES [23], have been suggested as an important step in harmonizing metabolic databases [24]. However, several tricky cases still remain unresolved. For example, InChI cannot handle the compound entries that contain R-groups.

Our neighborhood-specific graph coloring method can derive atom identifiers for every atom in a specific compound with consideration of molecular symmetry, facilitating the construction of an atom-resolved metabolic network [25]. Furthermore, a unique compound coloring identifier can be generated based on the atom identifiers, which can be used for compound harmonization across metabolic databases. The results derived from the compound coloring identifiers were quite promising. However, issues like incomplete atomistic details were not completely handled in that prior work.

To put this paper into context with our prior published work, we first developed the subgraph isomorphism detection algorithm CASS (Chemically Aware Substructure Search) in 2014 [26] and have made multiple improvements to this code base over the years and now call it BASS (Biochemically Aware Substructure Search). In developing our neighborhood-specific graph coloring method, we further enhanced BASS to efficiently detect aromatic substructures which was required for that work. In this paper, we further enhanced BASS to aid in the validation of generic compound pairs and to efficiently derive atom mappings from KEGG RDM descriptions. In addition, we solved inconsistent atomistic characteristics across databases by defining a set of harmonization relationship types between compounds, aiming to capture chemical details while maintain compound pairs at various levels. Furthermore, we used the classification of compound pairs and EC (Enzyme Commission) numbers to harmonize metabolic reactions across Kyoto Encyclopedia of Genes and Genomes (KEGG) and MetaCyc metabolic pathway databases via establishing hierarchical harmonization relationships between metabolic reactions. We further made use of the atom identifiers to evaluate atom mapping consistency of these harmonized reactions. Through this analysis, we detected some issues that cause the inconsistency of reaction atom mappings both within and across databases. The generalization of metabolic reactions can be applied to various interesting topics including but not limited to predicting biotransformation of newly discovered metabolites [27], devising novel synthetic pathways of essential metabolites [28], and bridging gaps in the current metabolic network [29]. Furthermore, expanding the existing metabolic network by integrating other metabolic databases can be easily achieved when the molfile representations [30] of compounds are provided. 

## 2. Results

### 2.1. Overview of KEGG and MetaCyc Databases

The compounds in the KEGG and MetaCyc databases are summarized in Table 1. Based on the atomic composition, we divided compounds into two groups: *specific compounds* (no R group) and *generic compounds* (with presence of R group(s)). About 8.02% KEGG compounds and 21.72% MetaCyc compounds contain R groups. 

According to the classification of compounds, we also categorized the atom-resolved metabolic reactions into two sets: specific reactions where all compounds in the reaction are specific compounds and generic reactions which contain at least one generic compound. Here, we only considered reactions with relatively complete EC numbers [31,32], since consistent EC number is one essential component in reaction harmonization. From Table 2, we can see that about 15% KEGG reactions and 34% MetaCyc reactions are generic reactions.

We further did a simple quality check of the atom-resolved reactions in KEGG and MetaCyc databases (Table 3). KEGG contains about 7.5% incomplete reactions where the number of atoms on both sides of the reaction is different. For MetaCyc, less than 0.5% reactions have incorrect atom mappings caused by mapping different atoms of different elements. In addition, a large amount of reactions only have part of atoms mapped in both KEGEG and MetaCyc databases. This level of incompleteness prevents their effective use in mass balanced metabolic modeling.

### 2.2. Results of Compound Harmonization across KEGG and MetaCyc Databases

With the loose compound coloring identifiers generated by the neighborhood-specific graph coloring method, about 8865 compound pairs were detected, including both generic and specific compound pairs [25]. However, some cases were not solved perfectly by the loose compound coloring identifiers. First, chemical details like atom and bond stereochemistry were ignored in the loose compound coloring identifies. Second, a compound pair can involve a generic compound and a specific compound (Figure 1), which cannot be discovered by the loosing compound coloring identifiers. The methyl group in KEGG compound C01042 can be a specification of the R group in MetaCyc compound CPD-576. What makes things more complicated is that a compound pair can be composed of two generic compounds with different atom composition. In Figure 2, even though both compounds contain an R group, the MetaCyc compound 3-Acyl-pyruvates can be regarded as a subgroup of compounds belonging to KEGG compound C00060. In addition, compound pairs with different structural representations, like tautomers, were missed by the loose compound coloring identifiers. 

#### 2.2.1. Harmonization of Specific Compounds

We first incorporated the chemical details, including atom stereochemistry and bond stereochemistry to evaluate the specific compound pairs detected by loose compound coloring identifiers. Incorporation of the chemical details can lead to three scenarios: (1) the paired compounds have the same set of chemical details; (2) the chemical details of one compound are the subset of the other compound; (3) the chemical details of the two compounds cannot be fully matched. Based on the above cases, we decided to classify the relationship between compound pairs as an equivalence relationship, a generic-specific relationship, or a loose relationship. With this classification, a compound in one database can be paired with multiple compounds in the other database with an appropriate relationship. With these improvements incorporated into specific compound harmonization (Table 4), we can see that the majority of specific compound pairs have a loose relationship, which is not surprising since the criteria for the loose relationship were less strict. Another explanation is that the chemical details for the same compound can be inconsistent across databases. The MetaCyc compound CPD-399 has a direct KEGG compound reference C03495 (Figure 3), but stereochemistry of some atoms in the two compound representations are not the same. 

#### 2.2.2. Harmonization of Generic Compounds

*Generic compounds* further complicate relationships between compounds. A generic compound can be related to generic and/or specific compounds (Figure 1 and Figure 2). For a compound pair of two *generic compounds* with the same atom composition, we classify them based on the same criteria of *specific compounds*. Harmonization of *generic compound* pairs of compounds with different chemical formulas is much more complicated, involving detection and validation steps. All chemical identifiers fail in detecting the possible pairs, including the loose compound coloring identifiers. On the other hand, it will be very time-consuming and unnecessary to do brute-force search of all compounds across databases. 

Here, we made use of the metabolic reactions across databases to detect the possible compound pairs with a different atom composition. We first extracted reaction pairs that can contain at least one *generic reaction* and share at least one EC number. Next, compounds with R group(s) in one reaction were paired with all the compounds in the other reaction. The validation method is described in the Materials and Methods section. Results of harmonization are summarized in Table 5. Most of the *generic compound* pairs have generic-specific relationships. This may be explained by the assumption that chemical details in a compound with less atoms are more likely to be included in the compound containing more atoms. 

#### 2.2.3. Harmonization of Compounds with Changeable Representations

Harmonization of compounds with changeable representation (e.g., linear vs. circular sugar representations) also requires detection and validation. Again, metabolic reactions were used to detect the possible compound pairs via an iterative approach (see Section 4.4). Two criteria should be obeyed when extracting the reaction pairs: (1) the two reactions should share at least one EC number; (2) at least a pair of compounds in the two reactions can be matched. For those unmatched compounds with the same chemical formula, they will be added to the possible list. The validation methods are described in the Materials and Methods section. About 45 such compound pairs were discovered after two rounds of iteration (Table 6). 

#### 2.2.4. Summary of Compound Harmonization

All compound pairs detected above were summarized in Table 7. In total, 15,704 compound pairs were discovered, and more than 80% of them were specific compound pairs, roughly in agreement with the proportion of generic compounds in the database. More importantly, about 2669 generic compound pairs were detected, which cannot be achieved by any existing chemical identifier. 

### 2.3. Results of Reaction Harmonization across KEGG and MetaCyc Databases

With the harmonized compounds, we performed reaction harmonization across KEGG and MetaCyc databases. Two criteria should be followed in reaction harmonization: (1) the two reactions should share at least an EC number; and (2) all compounds in the two reactions should be paired unless one reaction has an extra compound entity, like H^+^. Reaction pairs were further categorized into the following three relationship types based on the classification of their compound pairs: (1) equivalence relationship when a reaction pair included only equivalently paired compounds; (2) generic-specific relationship when a reaction pair only included equivalently paired compounds and at least one generic-specific compound pair that are consistently in the same general-to-specific direction; (3) loose relationship when a reaction pair included loosely paired compounds or generic-specific paired compounds with inconsistent general-to-specific direction. 

We first harmonized the specific metabolic reactions where both reactions are specific reaction (Table 8). We can see that reaction pairs in group 3 take up more than 70%, which is quite consistent with the classification of specific compound pairs. About 60% of specific compound pairs are loosely matched (Table 4), and a reaction pair only requires one loosely matched compound pair to be classified into group 3. 

We also analyzed the generic reaction pairs where at least one reaction is generic. Above 70% generic reaction pairs are in group 2 (Table 9), which can also be well explained by the previous result that around 95% generic compound pairs have a generic-specific relationship.

Since the EC information is not very complete in both databases, some reaction pairs can be ignored due to the mismatch or miss of the last level EC. To avoid missed pairs, we relaxed the first criterion in reaction harmonization to “the two reactions should have at least a pair of EC numbers that share the first 3 levels”. The newly discovered reaction pairs are summarized in Table 10, including both specific and generic reaction pairs. Either mismatch or miss of last EC occur in some reaction pairs. Specific examples are shown in Appendix A.

The results of reaction harmonization are shown in Table 11. Overall, 3856 reaction pairs were detected via EC numbers and integrated compound pairs. The majority of reaction pairs are specific. About 10% of reactions pairs can be missed due to incomplete and inconsistent EC numbers.

### 2.4. Comparison of KEGG RCLASS and RPAIR Data

For the KEGG database, the RCLASS data describes the chemical transformation of substrate-product in the RDM pattern [33]. A RDM description can be divided into three parts: reaction center (R), the different region (D), and the matched region (M). In order to distinguish functional groups and microenvironment of atoms, KEGG classified atomic species of C, N, O, S, and P into 68 types (KEGG atom types) [34], which are implemented in the RDM description. As shown in Figure 4, The RCLASS entry RC00003 contains one RDM description. The S atom is the reaction center, the C1a in the first substructure belongs to the different region, and those C1b atoms are in the matched region. Based on the RDM pattern, we derived the atom mappings for specific reactant-product compound pairs based on a common graph isomorphism search between the two compounds limited by RDM description. We successfully parsed atom mappings for 10,212 (out of 10,313) compound pairs. There are 76 compound pairs that cannot be deciphered due to the incorrect or missing descriptions of reaction centers (Appendix A). For complicated compound pairs with multiple reaction centers, each reaction center can be mapped to several different atoms, which in a few instances causes a serious combinatorial issue that is impossible to address in a reasonable amount of time. An example is shown in Figure 5. Roughly 10^13^ possible cases can be derived based on the RDM descriptions. In total, 25 compound pairs cannot be processed owing to this combinatorial problem (Appendix A). KEGG used to archive the atom mappings between the reactant-product compound pairs in the RPAIR database, where the mapped atoms are specified by the atom numbering for a compound pair. Here, we evaluated the atom mappings derived from RCLASS and an older version of KEGG RPAIR. The majority (great than 86%) of atom mappings between RCLASS and RPAIR are the same (Table 12). To further validate the results, we calculated the fraction of atom mappings with changed local bonded chemical environment across the mapping (i.e., atom mappings with changed one-bond atom color) in terms of the total number of mapped atoms in the reaction. The expectation is that this fraction represents the fraction of reaction center atoms present where a chemical bond is changed or broken. Then, we generated a scatter plot of changed local atom color fraction for KEGG RPAIR versus RCLASS atom mappings. From Figure 6A, we can see that the majority compound pairs have the same fraction of changed local atom color (concentrated on the diagonal line). In addition, more atom mappings derived from RPAIR have a higher ratio of changed atoms. We figured out that the majority of the inconsistency is due to the interchangeable mappings of resonant atoms, like the O atoms in the carboxyl group (Appendix A). After further curation to handle resonant atoms (Table 13), about 94% compound pairs have the same atom mappings. From Figure 6B, we can see that quite large portion of compound pairs with higher ratio of changed atoms in RPAIR disappear. The remaining 557 inconsistent mappings appear to come from two different issues. One, more than 93% of the remaining inconsistent mappings (517 out of 557) are likely caused by the updating of the KCF (molfile like) files or associated molfiles in KEGG database from continual curation. For example, the RDM description for compound pair C01255_C02378 has been updated in the RCLASS (Appendix A). We also plotted the changed one-bond atom color fraction for compound pairs with RDM update (Figure 6C). Compound pairs in either RCLASS or RPAIR can have higher changed atom ratio. The fraction of changed local atom color appears to equally distributed above and below the diagonal red line, which is interesting since the update of RDM descriptions is a correction process in KEGG database and may reflect both changes in specific mapped atoms and changes in the overall proportion of atoms mapped. Two, we found that a compound representation can vary across different compound pairs. Therefore, we hypothesize that a lack of synchronization between compound and compound_pair representations over time has caused the observed atom mapping inconsistencies detected in most of the other 40 compound pairs. For this part, RPAIR compound pairs show an increased fraction of changed local atom color (Figure 6D) versus its equivalent RCLASS, demonstrating that this metric has value in evaluating atom mappings.

### 2.5. Evaluation of Atom Mappings between KEGG and MetaCyc Databases

The atom mappings for each reaction in the MetaCyc database are specified based on the atom numbering of each compound in their molfile representation. For the KEGG database, we used the atom mappings for compound pairs parsed from the RCLASS entries. Here, we evaluated the atom mappings in about 3000 specific reaction pairs with the same compound representations (Table 14). About 88% of the reaction pairs have consistent atom mappings between the two databases. A consistent example is shown in Figure 7.

We also generated a scatter plot of changed atom color fraction between paired KEGG and MetaCyc reactions (Figure 8A). For some reactions, only part of the compounds are mapped in either database (Table 3). For MetaCyc, atoms are normally mapped at a reaction level. Since the KEGG RCLASS database maps atoms at a compound level, multiple RCLASS atom mappings must be evaluated together for a given KEGG reaction. We also just visualized paired reactions with inconsistent atom mappings (Figure 8B). We can see that the MetaCyc reactions have a higher ratio of changed local atom color. However, the number of mapped atoms in the paired reactions are not always the same, which can cause the fraction of changed local atom color can deviate from the diagonal. This issue makes a direct interpretation for specific reaction pairs more difficult, but the observed trend above the diagonal has interpretable value. 

Through these comparisons, we see that both databases can contain distinct issues with their atom mappings. Some MetaCyc reactions can have incorrect atom mappings. An example is shown in Figure 9. For some KEGG reactions with single compound involving in several compound pairs, one atom can be mapped to multiple atoms and leave some atoms unmatched. For the KEGG reaction R10579 shown in Figure 10A, based on the RDM descriptions in the two compound pairs (Figure 10B,C), atom 1 in compound C00251 is mapped to atom 1 in compound C00022 and atom 1 in compound C00578, leaving atom 2 in C00251 unmapped. Compared with the corresponding MetaCyc reaction RXN-14940 (Figure 11), the RDM description of KEGG RCLASS RC03212 appears incorrect. The harmonized reactions with different atom mappings are shown in Appendix A. 

## 3. Discussion

Effective integration of compound and reaction from various sources is hard to achieve due to incomplete and inconsistent atom-level and bond-level details, like R groups and stereochemistry, across databases. First, we categorized compounds into specific and generic compounds based on the presence of R groups. Meanwhile, metabolic reactions were classified into specific and generic reactions according to the presence of generic compounds. To overcome inconsistent atomistic characteristics, a set of relationships between compounds were defined to both keep chemical details and conserve compound pairs at various levels. According to the degree of consistency, compound pair relationships are classified into three types: equivalence, generic-specific, and loose relationships. The majority (around 60%) of specific compound pairs have loose relationships, confirming the inconsistent issues in the databases to some extent. To our knowledge, no chemical identifier can be used to directly harmonize generic compounds across databases. Here, we further optimized a subgraph isomorphism detection algorithm to validate generic compound pairs. We first made use of the metabolic reactions to discover possible generic compound pairs. After validation, 2669 generic compound pairs remained. In addition, we developed pragmatic methods to validate tautomers and compounds with linear and circular representations. We discovered 45 compound pairs of compounds with the same chemical formula but fundamentally different structures, for example linear versus circularized chemical representations. In total, 15,704 harmonized compound pairs were detected, which dwarfs our prior best published compound harmonization result of 8865 harmonized compound pairs and 5681 harmonized compound pairs identified by prior identifiers and methods. Next, we mapped atom-resolved metabolic reactions across KEGG and MetaCyc via compound pairs and EC numbers. Reaction pairs were also catalogued into hierarchical relationships in accordance with the classification of compound pairs. About 3856 harmonized reaction pairs were detected, and 10% of them can be missed by mismatched EC numbers (Appendix A), strongly suggesting that curation of EC numbers is of great importance in reaction harmonization. A prior systematic comparison of KEGG and MetaCyc had detected only 1961 shared reactions; however, this comparison was published in 2013 [15]. The BRaunschweig ENzyme Database (BRENDA) indicates in a 2019 paper that 6115 reactions are harmonizable between KEGG and MetaCyc [35]. However, BRENDA uses a combination of text mining and prediction algorithms to build their database from primary literature, likely making their harmonization results not as chemically specific as the results presented here which directly analyzes molfiles provides by KEGG and MetaCyc.

Furthermore, we made use of the atom identifiers derived from our neighborhood-specific graph coloring method to evaluate the consistency of atom mappings across harmonized reactions. About 88% of reaction pairs have consistent atom mappings. For the 12% of harmonized and comparable reactions that are inconsistent, we do not have ground truth for determining which version of the reaction is correct. However, the fraction of changed local atom color provides a uses metric for suggesting which version has higher confidence. Additionally, given that these reaction descriptions represent reactions across thousands of organisms, it is possible that both versions are correct in different organisms. Additionally, we determined that both databases contain issues leading to inconsistency. For example, atoms in some MetaCyc reactions are not mapped correctly. For KEGG, we detected unsynchronized atom numbering in the older KEGG RPAIR representation, which is likely the reason that KEGG removed RPAIRS from their public version of the database. In contrast, the KEGG RCLASS provides a concise RDM representation of reaction atom mappings between a reaction-product compound pair, which appears highly resistant to consistency errors. This resistance to consistency error is due to a decoupling of the atom mappings from the specific atom order in the molfile representations. This allows the molfile representations to be minorly updated without having to update the RDM descriptions. However, there are also some issues with RDM descriptions. About 76 compound pairs cannot be parsed due to the incorrect description of reaction centers, and parsed compound pairs can be unreasonable at reaction level. In addition, a few KEGG RCLASS entries are computationally difficult to decipher due to a combinatorial issue caused by the several factors: multiple reaction centers in a single reaction, symmetric compounds, and reaction descriptions involving multistep reactions. This combination of factors introduces a large number of possibilities with matching a list of RDM descriptions to specific reaction center atoms. One way to prevent this combinatorial problem is to represent multiple reaction center atoms with their associated difference atoms and match atoms within a paired substructure representation instead of a list of RDM descriptions. Figure 10B illustrates this paired substructure representation for the KEGG RCLASS RC02148. This kind of paired RDM substructure representation would allow the use of an efficient subgraph isomorphism detection method to derive the atom mappings and could be represented as a pair of molfiles along with a mapping of atoms between the two molfiles, all stored within a single sdfile. Additionally, our compound harmonization method for harmonizing changeable compound pairs would be useful for updating the paired RDM substructures when the compound representations dramatically change. 

In addition, the methods we developed can be easily applied to integrate other metabolic databases that provide molfile representations of compounds, facilitating the expansion of the existing metabolic networks. Moreover, this hierarchical framework for relating compounds and reactions is a possible first step towards creating a systematic organization of all reaction descriptions at a desired chemical specificity to fit a given application. Such a systematic organization of reaction descriptions would augment the current Enzyme Commission number system and be useful to a wide range of possible applications from metabolic modeling, metabolite and reaction prediction, and network incorporation of newly discovered metabolites. 

## 4. Materials and Methods

### 4.1. Compound and Metabolic Reaction Data

All data were downloaded directly from KEGG (https://www.genome.jp/kegg/ accessed on 1 April 2021) and MetaCyc (https://metacyc.org/ accessed on 1 April 2021) databases. MetaCyc compound and reaction data downloaded from BioCyc is in version 23.0. The KEGG COMPOUND, KEGG REACTION and KEGG RCLASS data is from the version available from KEGG on April 2021 via its REST interface. KEGG RPAIR data was downloaded from KEGG database in 2016.

### 4.2. Curation of Molfile

The documentation of atom stereochemistry in the molfiles is not complete. We used Open Babel [36] to curate the original molfiles and add stereospecific information.

### 4.3. Identification of Double Bond Stereochemistry

We previously adopted a method for automated identification of double bond stereochemistry [37]. One limitation of this method is that only double bonds between two carbon atoms can be handled. For example, double bonds connected by heterogenous atoms, like N=C, cannot be processed by the method. Here, we designed a new algorithm to distinguish cis/trans stereoisomers. The same criteria are applied to assign priority to each group attached to the double bond. If one side of the double bond only has one group, this group will be prioritized. Next, the 2D plane of the compound representation is divided into two parts with line crossing the double bond. If the prioritized groups of both sides are on the same part of the divided plane, the double bond is labeled as cis; otherwise, it is trans.

### 4.4. Flowchart of Steps in the Compound and Reaction Harmonization Process

The flowchart of steps in compound and reaction harmonization is shown in Figure 12. The initial compound pair list is composed of compound pairs detected by the loose compound coloring identifiers. Next, reaction harmonization is conducted with the compound pair list. Two criteria are obeyed in reaction harmonization: the two reactions should share at least one EC number and all compounds in the two reactions are paired unless one reaction has an extra compound entity, like H^+^. Apart from valid reaction pairs, reaction pairs with the same EC number and some unmatched compounds are also extracted. We hypothesized that those unmatched compounds are likely to be compound pairs. Validation is conducted for the unmatched compounds, and the valid compound pairs are added to the compound pair list. Every time the compound pair list is updated, the above process is repeated until no new compound pairs are discovered.

### 4.5. Validation of Tautomers

Most common form of tautomerization involves a hydrogen changing places with a double bond. Based on this transformation, the following steps are performed to validate if two compounds with same chemical formula are tatutomers. To eliminate the difference caused by single and double bonds in the structural representation, all the double bonds are converted into single bonds, and the subgraph isomorphism detection algorithm [26] is used to check if two structural representations are the same after modification. Next, double bonds at unmatched positions are examined. If all the mismatches are caused by possible tautomerization, the compound pair is considered valid. Finally, other chemical details not related to atoms in the changeable positions are compared to classify the relationship between valid pairs. 

### 4.6. Validation of Generic Compound Pairs of Compounds with Different Chemical Formula

For two compounds A and B, the subgraph isomorphism detection algorithm [26] is used to verify if the graph representation of A (ignoring R and H) is contained in the graph representation of B. Then, each unmatched branch in B is examined if it corresponds to an R group in A. Compound pairs that meet both criteria are considered valid. Next, the chemical details (atom and bond stereochemistry) in the two compounds are compared for relationship type classification. If the chemical details of compound A are included in compound B, then A has a generic-specific relationship to B; otherwise, A and B have a loose relationship. 

### 4.7. Validation of Compound Pairs with Linear and Circular Representations

The compound with changeable linear and circular structures are common in small molecule carbohydrate metabolites, like glucose. This conversion occurs due to the ability of aldehydes and ketones to react with alcohols. To validate the compound pairs with linear and circular representations, we first locate the bond in the circular structure that is formed by connecting the C in the aldehyde (keto) group and O in the hydroxy group. The following steps include breaking the newly formed bond and restoring the C=O bond in the aldehyde (keto) group. Then, a new compound coloring identifier is generated for the modified circular representation. If the updated compound coloring identifiers match, the compound pair is considered valid. 

### 4.8. Parse of KEGG RCLASS RDM Patterns

Based on the RDM patterns, we first identified the possible atoms that can be mapped to each reaction center. Then, we derived the possible combinations of atoms for all the reaction centers for each compound. We paired cases in either compound, removed changed bond in the compound according to different region, and detected the maximum common subgraph of the remaining structures. We examined all the combinations and derived the optimal mappings with the maximum number of mapped atoms and least ratio of changed atoms.

## Figures and Tables

**Figure 1 metabolites-11-00431-f001:**
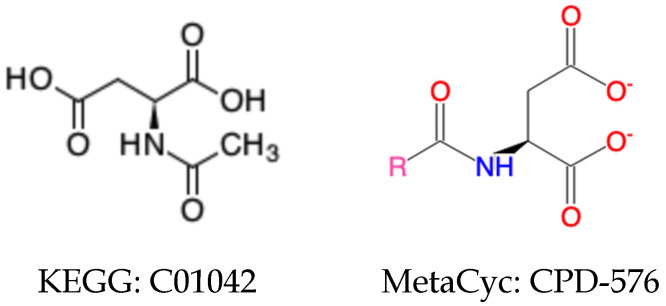
Compound pair of generic and specific compounds.

**Figure 2 metabolites-11-00431-f002:**
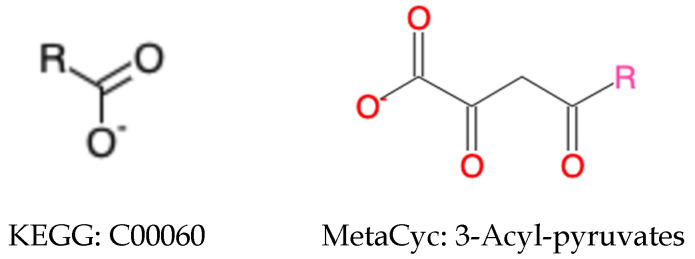
Compound pair of generic compounds.

**Figure 3 metabolites-11-00431-f003:**
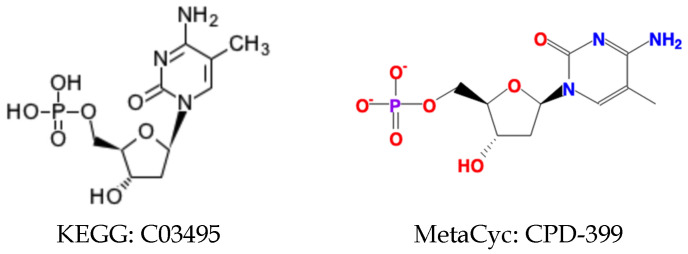
Example harmonized compound pair of compounds with inconsistent chemical details.

**Figure 4 metabolites-11-00431-f004:**
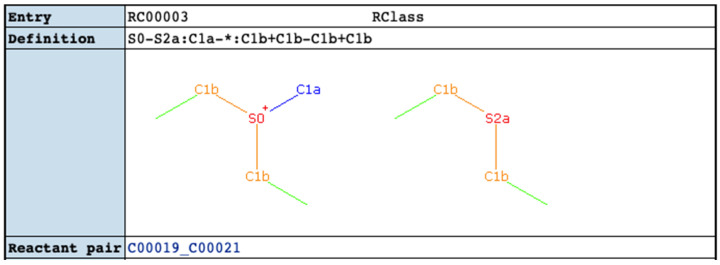
Example of RCLASS entry (https://www.genome.jp/dbget-bin/www_bget?rc:RC00003 accessed on 1 April 2021).

**Figure 5 metabolites-11-00431-f005:**
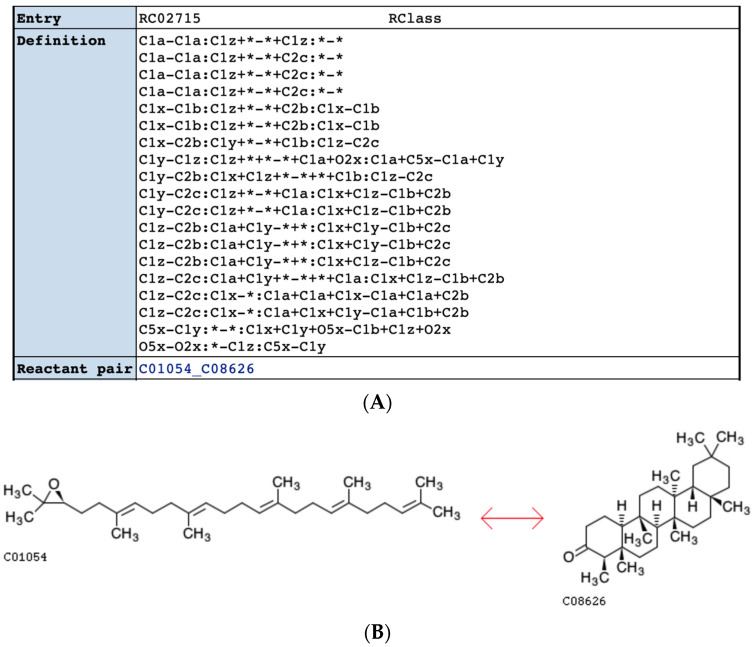
RCLASS with combinatorial issue caused by multiple possible mappings of RDM descriptions. (**A**) RDM description of RCLASS RC02715 (https://www.genome.jp/dbget-bin/www_bget?rc:RC02715 accessed on 1 April 2021); (**B**) Compound pair C01054_C08626 that follows the RC02715 RDM pattern (https://www.genome.jp/Fig/reaction/R09910.gif accessed on 1 April 2021).

**Figure 6 metabolites-11-00431-f006:**
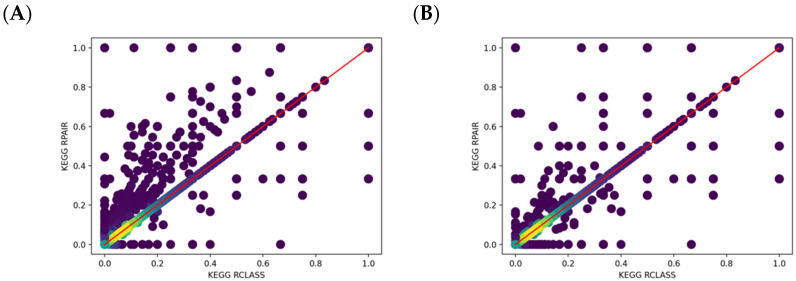
Scatter plots of changed one-bond atom color fraction for KEGG RCLASS versus RPAIR atom mappings in paired compounds. Lighter colors represent higher overlapped point density. (**A**) All compound pairs before correcting resonant atoms; (**B**) All compound pairs after correcting resonant atoms; (**C**) Compound pairs with inconsistent atom mappings caused by RDM update; (**D**) Compound pairs with inconsistent atom mappings caused by unsynchronized representations.

**Figure 7 metabolites-11-00431-f007:**
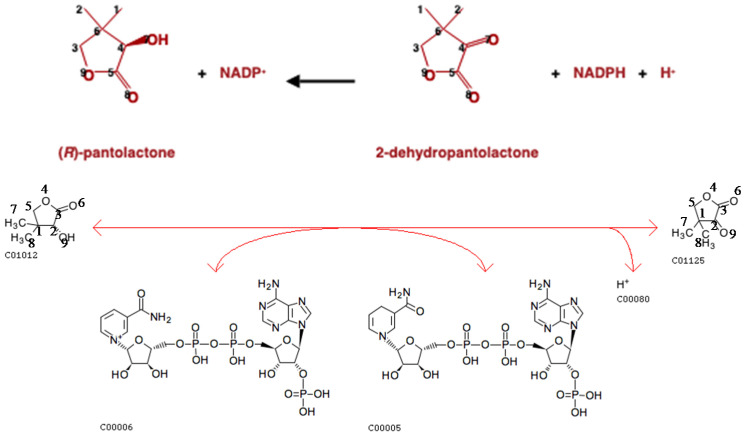
Reaction pair MetaCyc 1.1.1.168-RXN (**top**, https://metacyc.org/META/NEW-IMAGE?object=1.1.1.168-RXN&&redirect=T accessed on 1 April 2021) and KEGG R03155 (**bottom**, https://www.genome.jp/entry/R03155 accessed on 1 April 2021) with the same atom mappings.

**Figure 8 metabolites-11-00431-f008:**
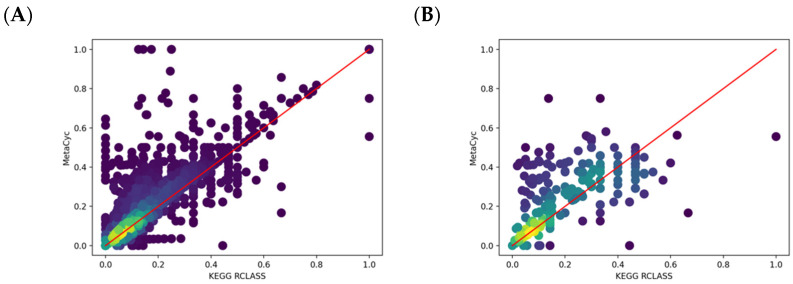
Scatter plots of changed one-bond atom color fraction for KEGG versus MetaCyc atom mappings in paired reactions. Lighter colors represent higher overlapped point density. (**A**) All paired reactions are included; (**B**) Reaction pairs with inconsistent atom mappings are included.

**Figure 9 metabolites-11-00431-f009:**
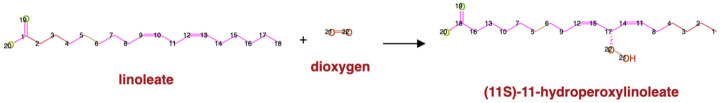
Example of MetaCyc reaction with incorrect atom mappings (https://metacyc.org/META/NEW-IMAGE?object=1.13.11.45-RXN&&redirect=T accessed on 1 April 2021).

**Figure 10 metabolites-11-00431-f010:**
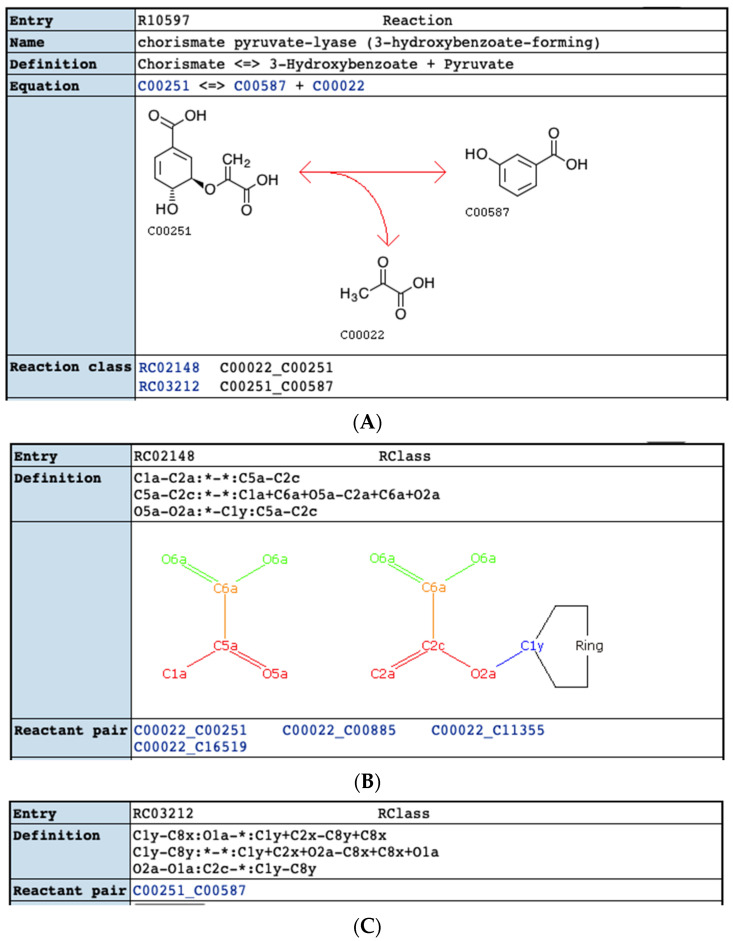
Example of KEGG reaction with incorrect mappings. (**A**) KEGG reaction R10579 (https://www.kegg.jp/entry/R10597 accessed on 1 April 2021); (**B**) KEGG RCLASS RC02148 (https://www.kegg.jp/entry/RC02148 accessed on 1 April 2021); (**C**) KEGG RCLASS 03212 (https://www.kegg.jp/entry/RC03212 accessed on 1 April 2021).

**Figure 11 metabolites-11-00431-f011:**
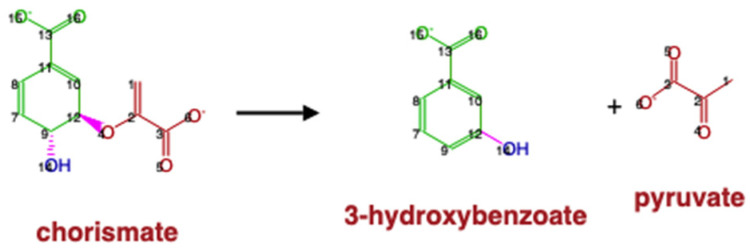
Atom mappings of MetaCyc reaction RXN-14940 (https://metacyc.org/META/NEW-IMAGE?object=RXN-14940&&redirect=T accessed on 1 April 2021).

**Figure 12 metabolites-11-00431-f012:**
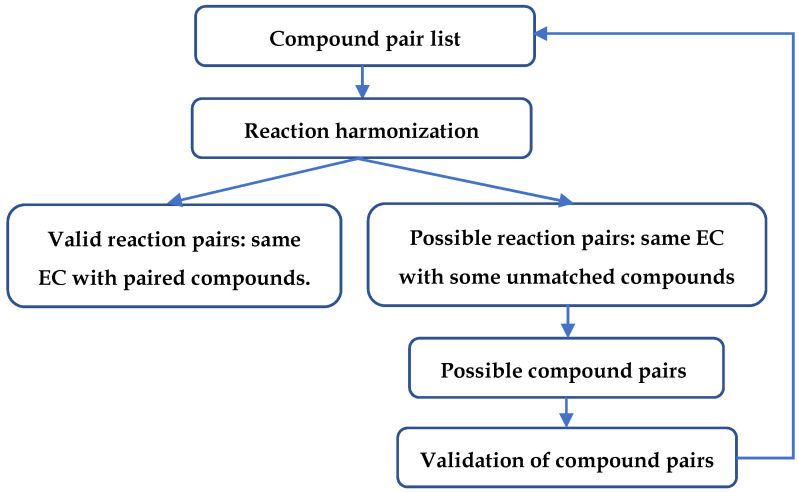
Flowchart of compound and reaction harmonization.

**Table 1 metabolites-11-00431-t001:** Summary of KEGG and MetaCyc compound databases.

Compound Type	KEGG	MetaCyc
specific compounds	16,529 (91.98%)	15,859 (78.28%)
generic compounds	1441 (8.02%)	4400 (21.72%)
Total	17,970 (100%)	20,259 (100%)

**Table 2 metabolites-11-00431-t002:** Summary of KEGG and MetaCyc atom-resolved metabolic reaction databases.

Reaction Type	KEGG	MetaCyc
specific reactions (4-leveled EC)	6780 (75.26%)	6397 (49.93%)
specific reactions (3-leveled EC)	886 (9.83%)	2022 (15.78%)
generic reactions (4-leveled EC)	1244 (13.81%)	3572 (27.88%)
generic reactions (3-leveled EC)	99 (1.10%)	822 (6.42%)
Total	9009 (100%)	12,813 (100%)

**Table 3 metabolites-11-00431-t003:** Quality check of atom-resolved reactions in KEGG and MetaCyc.

Database	Incomplete Reaction	Incorrect Atom Mappings	Incomplete Atom Mappings
KEGG	772 (7.53%)	0	7213 (70.36%)
MetaCyc	0	54 (0.37%)	6130 (41.87%)

**Table 4 metabolites-11-00431-t004:** Harmonization of specific compound pairs.

Relationship Type	Count
equivalence	3636 (27.99%)
generic-specific	1712 (13.18%)
loose	7642 (58.83%)
Total	12,990 (100%)

**Table 5 metabolites-11-00431-t005:** Harmonization of generic compounds.

Relationship Type	Count
equivalence	126 (4.72%)
generic-specific	2543 (95.28%)
loose	0
Total	2669 (100%)

**Table 6 metabolites-11-00431-t006:** Harmonization of compounds with changeable representations.

Relationship Type	Count
equivalence	20 (44.44%)
generic-specific	0
loose	25 (55.56%)
Total	45 (100%)

**Table 7 metabolites-11-00431-t007:** Summary of compound harmonization between KEGG and MetaCyc.

Compound Pair Type	Count
specific compound pairs	12,990 (82.72%)
generic compound pairs	2669 (16.99%)
changeable compound pairs	45 (0.29%)
Total	15,704 (100%)

**Table 8 metabolites-11-00431-t008:** Harmonization of specific reactions between KEGG and MetaCyc.

Relationship Type	Count
equivalence	718 (24.00%)
generic-specific	68 (2.27%)
loose	2205 (73.72%)
Total	2991 (100%)

**Table 9 metabolites-11-00431-t009:** Harmonization of generic reactions between KEGG and MetaCyc.

Relationship Type	Count
equivalence	29 (6.03%)
generic-specific	344 (71.51%)
loose	108 (22.45%)
Total	481 (100%)

**Table 10 metabolites-11-00431-t010:** Loose harmonization of reactions between KEGG and MetaCyc.

Relationship Type	Count
equivalence	49 (12.76%)
generic-specific	96 (25.00%)
loose	239 (62.24%)
Total	384 (100%)

**Table 11 metabolites-11-00431-t011:** Summary of reaction harmonization between KEGG and MetaCyc.

Reaction Pair Type	Count
specific	2991 (77.57%)
generic	481 (12.47%)
loose EC	384 (9.96%)
Total	3856 (100%)

**Table 12 metabolites-11-00431-t012:** First-round evaluation of atom mappings of compound pairs between KEGG RCLASS and RPAIR.

Condition	Count
same atom mappings	8017 (86.1%)
inconsistent atom mappings	1294 (13.9%)
Total	9311 (100%)

**Table 13 metabolites-11-00431-t013:** Second-round evaluation of atom mappings of compound pairs between KEGG RCLASS and RPAIR.

Condition	Count
same atom mappings	8754 (94.02%)
inconsistent atom mappings	557 (5.98%)
Total	9311 (100%)

**Table 14 metabolites-11-00431-t014:** Evaluation of atom mappings between KEGG and MetaCyc.

Condition	Count
same atom mappings	2685 (88.0%)
inconsistent atom mappings	366 (12.0%)
Total	3051 (100%)

## Data Availability

All data used and the results generated in the manuscript are available on: https://doi.org/10.6084/m9.figshare.14703999 accessed on 1 April 2021.

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
