# Peer review of "Hierarchical Harmonization of Atom-Resolved Metabolic Reactions across Metabolic Databases"

_metabolites, 2021, doi:10.3390/metabo11070431_

Round 1

Reviewer 1 Report

The work presented by authors is motivated by the need “to harmonize compounds and reactions across different metabolic databases” in order to make reliable use of metabolic pathways.  As stated in the abstract by the authors: “Here, we optimized a subgraph isomorphism detection algorithm to validate generic compound pairs. “  The manuscript follows earlier work (Metabolites. 2020 Sep 11), in which the authors developed “a neighborhood-specific graph coloring method that creates unique identifiers for each atom in a compound facilitating construction of an atom-resolved metabolic network”. The methodology employed was based on two independent detection methods: Biochemically Aware Substructure Search (BASS) method using neighborhood-specific graph coloring, and the aromatic detection facilities in the Indigo package.  Another earlier manuscript (CASS for Chemically Aware Sub-structure Search) automatically detected functional groups in compound libraries.

In earlier published work, authors have indicated that the methodology presented “greatly improve subgraph isomorphism detection”.  Therefore, the present manuscript is a newly optimized subgraph isomorphism detection that is subsequently applied to the databases mentioned in the manuscript. 

Although a number of consistency issues are reported, the significance of the reported results is difficult to evaluate because the ground truth used for evaluation is not made clear.  It may be the case that a ground truth is difficult to define; then, it would be useful for the authors to help readers understand the significance of the narrative presented (beyond simply stating the outcome of a computation).  If, on the other hand, the ground truth is known, then the criteria for evaluation against the ground truth and the ground truth need to be clearly stated.

An optimization addressing the subgraph isomorphism is worthy of publication by itself.  However, the proposed optimization mentioned in the abstract (and in the body) is not explained further.  Consulting the authors’ software page on the web, the software that is mentioned in this manuscript and available on the web site appears to be an older version and perhaps lacking the proposed optimization noted in the abstract.  The supplementary material submitted does not contain the needed information either. Therefore, the nature of the optimization method and its significance is hard to determine.

The manuscript addresses an important problem.  It would benefit from a clearer description of the significance of the results (in the context of the ground truth, as mentioned earlier).  In addition, an algorithm and clear explanation of the "optimization" as used in this latest software (and utilized for the results of this manuscript) would contribute greatly to the significance of the work presented.

Author Response

Reviewer 1:

The work presented by authors is motivated by the need “to harmonize compounds and reactions across different metabolic databases” in order to make reliable use of metabolic pathways.  As stated in the abstract by the authors: “Here, we optimized a subgraph isomorphism detection algorithm to validate generic compound pairs. “  The manuscript follows earlier work (Metabolites. 2020 Sep 11), in which the authors developed “a neighborhood-specific graph coloring method that creates unique identifiers for each atom in a compound facilitating construction of an atom-resolved metabolic network”. The methodology employed was based on two independent detection methods: Biochemically Aware Substructure Search (BASS) method using neighborhood-specific graph coloring, and the aromatic detection facilities in the Indigo package.  Another earlier manuscript (CASS for Chemically Aware Sub-structure Search) automatically detected functional groups in compound libraries.

In earlier published work, authors have indicated that the methodology presented “greatly improve subgraph isomorphism detection”.  Therefore, the present manuscript is a newly optimized subgraph isomorphism detection that is subsequently applied to the databases mentioned in the manuscript.

Response:

We thank the reviewer for their efforts in reviewing our manuscript.  But the reviewer’s synopsis of our manuscript and prior publications is not completely accurate.  To clarify, we developed and published CASS back in 2014 and applied it to a chemical functional group detection to evaluate strategies for chemoselective probes in metabolite identification.  Over the years, we have made major improvements to this code base for other purposes and applications.  Our Metabolites 2020 Sep 11 publication showed a major improvement in CASS (which we now call BASS due to a range of improvements) used in aromatic group detection which was needed to implement the neighborhood-specific graph coloring method and compound harmonization.  In the current manuscript, we made further improvements to both BASS to enable derivation of atom mappings from KEGG RDM descriptions.  We also made large improvements to our atom coloring method and then developed combined compound and reaction harmonization methods that nearly doubled the number of harmonized compounds from the previous 2020 publication as well as provided reaction harmonization.  To make this progression clearer, we have added the following to the manuscript:

“To put this paper into context with our prior published work, we first developed the subgraph isomorphism detection algorithm CASS (Chemically Aware Substructure Search)  in 2014 [26] and have made multiple improvements to this code base over the years and now call it BASS (Biochemically Aware Substructure Search).  In developing our neighborhood-specific graph coloring method, we further enhanced BASS to efficiently detect aromatic substructures which was required for that work.  In this paper, we further enhanced BASS to aid in the validation of generic compound pairs and to efficiently derive atom mappings from KEGG RDM descriptions…”

Issue 1:

Although a number of consistency issues are reported, the significance of the reported results is difficult to evaluate because the ground truth used for evaluation is not made clear.  It may be the case that a ground truth is difficult to define; then, it would be useful for the authors to help readers understand the significance of the narrative presented (beyond simply stating the outcome of a computation).  If, on the other hand, the ground truth is known, then the criteria for evaluation against the ground truth and the ground truth need to be clearly stated.

Response:

The reviewer is correct.  The evaluation of atom mappings for harmonized reactions is difficult, since we do not have ground truth; however, we have provided a useful metric for evaluation which is the fraction of changed local atom color.  We have made this point in the discussion:

“For the 11% of harmonized and comparable reactions that are inconsistent, we do not have ground truth for determining which version of the reaction is correct.  However, the fraction of changed local atom color provides a uses metric for suggesting which version has higher confidence.  Also, given that these reaction descriptions represent reactions across thousands of organisms, it is possible that both versions are correct in different organisms.”

Also, this line of thinking led us to add Table 3 that provides a summary of issues in KEGG and MetaCyc reaction descriptions, along with S Table 1 that lists KEGG RCLASS entries that have incorrect RDM descriptions that complements S Table 2 that lists KEGG RCLASS entries with compound pairs that are computationally intractable. Furthermore, we expanded Figure 6 to four scatterplots which highlights two types of inconsistencies detected between RPAIR and RCLASS entries.  We also expanded Figure 8 to make it easier to see trends in the fraction of changed local atom color between KEGG and MetaCyc reactions and better described the caveats in their interpretation. 

Issue 2:

An optimization addressing the subgraph isomorphism is worthy of publication by itself.  However, the proposed optimization mentioned in the abstract (and in the body) is not explained further.  Consulting the authors’ software page on the web, the software that is mentioned in this manuscript and available on the web site appears to be an older version and perhaps lacking the proposed optimization noted in the abstract.  The supplementary material submitted does not contain the needed information either. Therefore, the nature of the optimization method and its significance is hard to determine.

Response:

The current codebase is a prototype and as such is provided “as is” in the FigShare repository (https://doi.org/10.6084/m9.figshare.14703999) that is described in the manuscript along with all of the results generated.  The original CASS codebase comes from before we started putting high quality codebases on GitHub: https://github.com/MoseleyBioinformaticsLab  and we do not put codebases under our GitHub organizational account until they are fully tested, documented, and distributable, typically via the PyPI, CRAN, or Bioconductor.  We do plan on making BASS available via GitHub and PyPI sometime NOT in the distant future.

Issue 3:

The manuscript addresses an important problem.  It would benefit from a clearer description of the significance of the results (in the context of the ground truth, as mentioned earlier).  In addition, an algorithm and clear explanation of the “optimization” as used in this latest software (and utilized for the results of this manuscript) would contribute greatly to the significance of the work presented.

Response:

As mentioned in our responses to Issue 1 and 2, we have added a ground truth context and the BASS and whole reaction harmonization codebases are available in a FigShare repository specifically created for this manuscript.  We are planning a later manuscript to present BASS with all of its’ enhancements, which is the appropriate venue for describing the specific details on that algorithm and implementation.

Reviewer 2 Report

In general, the article is interesting, simple and effective methods of comparing two databases are considered. All data and code in Python are open source, which is nice. It might make sense to combine at least some of the tables.This amount does not improve understanding.

Perhaps it is worth giving data on how much the “pairings” established in this research overlap with direct links in MetaCyc to KEGG.

Line 245: typo - correct RC00003 - with four zeros.

Author Response

Reviewer 2:

In general, the article is interesting, simple and effective methods of comparing two databases are considered. All data and code in Python are open source, which is nice. It might make sense to combine at least some of the tables.This amount does not improve understanding.

Response:

We thank the reviewer for their kind words and recognition of our methods.  As far as the number of tables, it is hard to make it clear in the text what results are referred to, if tables are combined.  It is much easier with Figures, since they can have multiple parts referred to with A,B,C,etc.

Issue 1:

Perhaps it is worth giving data on how much the “pairings” established in this research overlap with direct links in MetaCyc to KEGG.

Response:

We already have published this in our 2020 atom coloring and compound harmonization paper where the 8,865 compound pairs detected were mostly a superset (5451 pairs as indicated in Table 5 of the prior paper) of the compound pairs identified by prior identifiers (5681 pairs as indicated in Table 2 in the prior paper).  This evaluation also included InChI strings we uniformly generated from KEGG and MetaCyc molfiles, which goes beyond identifiers directly provided by KEGG and MetaCyc. However, it is important to put the current results in the context of prior work.  Therefore, we have added the following to the discussion:

“In total, 15,704 harmonized compound pairs were detected, which dwarfs our prior best published compound harmonization result of 8,865 harmonized compound pairs and 5,681 harmonized compound pairs identified by prior identifiers and methods.”

Issue 2:

Line 245: typo - correct RC00003 - with four zeros.

Response:

Thanks! Fixed.

Reviewer 3 Report

This paper describes the development of an optimized subgraph isomorphism detection algorithm to validate generic compound pairs. In addition, the authors have defined a set of harmonization relationship types between compounds to deal with inconsistent chemical details. The results demonstrated that 15,704 compound pairs across KEGG and MetaCyc databases were detected. Using the classification of compound pairs and EC (Enzyme Commission) numbers of reactions, the authors achieved  hierarchical relationships between metabolic reactions. The work is interesting, since metabolic models of a wide range of organisms are very useful tools in system biology.

Comments

1) Materials and methods: for all used databases the authors should add their web addresses.  

2) The authors need to emphasize in the introduction the usefulness of the algorithm CASS and its application.

3) It would be useful the authors to consider adding some more recent citations (there are only a few  recent references in the lists (2018-2020).

4) A conclusion section would be useful to point out the main outcomes of the study.

Author Response

Reviewer 3:

This paper describes the development of an optimized subgraph isomorphism detection algorithm to validate generic compound pairs. In addition, the authors have defined a set of harmonization relationship types between compounds to deal with inconsistent chemical details. The results demonstrated that 15,704 compound pairs across KEGG and MetaCyc databases were detected. Using the classification of compound pairs and EC (Enzyme Commission) numbers of reactions, the authors achieved  hierarchical relationships between metabolic reactions. The work is interesting, since metabolic models of a wide range of organisms are very useful tools in system biology.

Response:

We thank the reviewer for their kind words and recognition of our methods and their significance. 

Issue 1:

Comments

1) Materials and methods: for all used databases the authors should add their web addresses. 

Response:

We have added this:

“All data were downloaded directly from KEGG (https://www.genome.jp/kegg/) and MetaCyc (https://metacyc.org/) databases.”

Issue 2:

2) The authors need to emphasize in the introduction the usefulness of the algorithm CASS and its application.

Response:

We have added the following which describes the evolution of CASS and BASS to handle a variety of common subgraph isomorphism problems:

“To put this paper into context with our prior published work, we first developed the subgraph isomorphism detection algorithm CASS (Chemically Aware Substructure Search)  in 2014 [26] and have made multiple improvements to this code base over the years and now call it BASS (Biochemically Aware Substructure Search).  In developing our neighborhood-specific graph coloring method, we further enhanced BASS to efficiently detect aromatic substructures which was required for that work.  In this paper, we further enhanced BASS to aid in the validation of generic compound pairs and to efficiently derive atom mappings from KEGG RDM descriptions…”

Issue 3:

3) It would be useful the authors to consider adding some more recent citations (there are only a few  recent references in the lists (2018-2020).

Response:

Unfortunately, we have not found many current references that add to this manuscript.  However, after a further search, we have found a couple new references and have explained our results in their context.  At least one of them is a 2019 reference.

“A prior systematic comparison of KEGG and MetaCyc had detected only 1,961 shared reactions; however, this comparison was published in 2013 [15]. The BRaun-schweig ENzyme Database (BRENDA) indicates in a 2019 paper that 6,115 reactions are harmonizable between KEGG and MetaCyc [35]. However, BRENDA uses a combination of text mining and prediction algorithms to build their database from primary literature, likely making their harmonization results not as chemically specific as the results presented here which directly analyzes molfiles provides by KEGG and MetaCyc.”

Issue 4:

4) A conclusion section would be useful to point out the main outcomes of the study.

Response:

The discussion acts as a detailed conclusions in this manuscript. While not concise, it does prevent misinterpretation of the main outcomes that can come from further reduction. In this instance, we feel that the possibility of misinterpretation outweighs the potential benefits from adding a conclusion section.

Round 2

Reviewer 1 Report

The revision by the authors have addressed my comments.